# Performance Assessment of Thermal Infrared Cameras of Different Resolutions to Estimate Tree Water Status from Two Cherry Cultivars: An Alternative to Midday Stem Water Potential and Stomatal Conductance

**DOI:** 10.3390/s20123596

**Published:** 2020-06-25

**Authors:** Marcos Carrasco-Benavides, Javiera Antunez-Quilobrán, Antonella Baffico-Hernández, Carlos Ávila-Sánchez, Samuel Ortega-Farías, Sergio Espinoza, John Gajardo, Marco Mora, Sigfredo Fuentes

**Affiliations:** 1Departamento de Ciencias Agrarias, Facultad de Ciencias Agrarias y Forestales, Universidad Católica del Maule, Curicó 3340000, Chile; javiera.antunez@alu.ucm.cl (J.A.-Q.); antonella.baffico@alu.ucm.cl (A.B.-H.); 2Research and Extension Center for Irrigation and Agroclimatology (CITRA) and Research Program on Adaptation of Agriculture to Climate Change (A2C2), Faculty of Agricultural Sciences, Universidad de Talca, Talca 3460000, Chile; carlos.avila@utalca.cl (C.Á.-S.); sortega@utalca.cl (S.O.-F.); 3Programa de Magíster en Hortofruticultura, Universidad de Talca, Talca 3460000, Chile; 4Departamento de Ciencias Forestales, Facultad de Ciencias Agrarias y Forestales, Universidad Católica del Maule, Talca 3460000, Chile; espinoza@ucm.cl; 5Instituto de Bosques y Sociedad, Facultad de Ciencias Forestales y Recursos Naturales, Universidad Austral de Chile, Campus Isla Teja, Valdivia 5090000, Chile; john.gajardo@uach.cl; 6Laboratory of Technological Research in Pattern Recognition (LITRP), Faculty of Engineering Science, Universidad Católica del Maule, Talca 3480112, Chile; mmora@ucm.cl; 7Digital Agriculture, Food and Wine Group, School of Agriculture and Food, Faculty of Veterinary and Agricultural Sciences, University of Melbourne, Parkville, VIC 3010, Australia; sigfredo.fuentes@unimelb.edu.au

**Keywords:** water relations, infrared thermography, camera resolution, crop water stress index, computer vision

## Abstract

The midday stem water potential (Ψs) and stomatal conductance (gs) have been traditionally used to monitor the water status of cherry trees (*Prunus avium* L.). Due to the complexity of direct measurement, the use of infrared thermography has been proposed as an alternative. This study compares Ψs and gs against crop water stress indexes (CWSI) calculated from thermal infrared (TIR) data from high-resolution (HR) and low-resolution (LR) cameras for two cherry tree cultivars: ‘Regina’ and ‘Sweetheart’. For this purpose, a water stress–recovery cycle experiment was carried out at the post-harvest period in a commercial drip-irrigated cherry tree orchard under three irrigation treatments based on Ψs levels. The water status of trees was measured weekly using Ψs, gs, and compared to CWSIs, computed from both thermal cameras. Results showed that the accuracy in the estimation of CWSIs was not statistically significant when comparing both cameras for the representation of Ψs and gs in both cultivars. The performance of all evaluated physiological indicators presented similar trends for both cultivars, and the averaged differences between CWSI’s from both cameras were 11 ± 0.27%. However, these CWSI’s were not able to detect differences among irrigation treatments as compared to Ψs and gs.

## 1. Introduction

Sweet cherry (*Prunus avium* L.) is a high-value fruit in the Southern Hemisphere fruticulture since they are of high demand fresh fruits in Northern Hemisphere markets [1]. The Mediterranean semi-arid agroclimatic conditions of the Central-South zone of Chile are optimal for cherry growth and production since the climate is cold in winter and dry with almost no rain events during the spring–summer seasons. These conditions allow the growth of semi-late- and late-maturing cultivars such as ‘Regina’ and ‘Sweetheart’, which present differences in the vegetative habit and vigor of trees [1,2]. This lack of rain during the growing season requires the adoption of technified irrigation, such as drip irrigation, for sweet cherry production to be commercially viable [3]. Considering the current climate change scenario, where the natural supply of water has decreased, investments in technified irrigation, such as micro-sprinkler or drip irrigation systems, are required by fruit growers to face water scarcity, and to precisely control irrigation scheduling [4,5]. Currently, fruit growers are applying irrigation strategies to increase water use efficiency (WUE) and water productivity (WP), such as regulated deficit irrigation (RDI) to reduce the amount of water irrigated during the growing season and obtain certain benefits such as improving water productivity [3,4,5]. The application of any RDI strategy requires precision in monitoring trees to avoid any detriment to the yield of fruit trees and to obtain maximum WP benefits. The tree monitoring of water stress has been traditionally performed using plant-based water stress indicators, where the midday stem water potential (Ψs) is one of the most used in commercial orchards [6]. This physiological parameter has been recommended as one of the most accurate tools for irrigation management [6,7]. The use of combined soil moisture sensors with the pressure chamber to measure Ψs have been suggested as reliable tools to help decision making for irrigation scheduling in fruit trees and grapevines [5]. Cherry trees are very sensitive to mild-to-severe water stress during the fruit growing cycle, which could negatively affect the yield and quality of fruit. Thus, the application of any kind of RDI strategies in these fruit trees is only recommended during the post-harvest period [3,8,9,10,11]. Examples of the application of Ψs for managing the irrigation of ‘Summit’ [10], ‘Prime Giant’ [8], and ‘Sweetheart’ sweet cherry trees [12], showed that a moderate post-harvest RDI treatment does not produce any detrimental effect on fruit yield and quality, substantially reducing the total water irrigated. Another sensitive physiological parameter to water stress is canopy conductance (gc), which represents the average leaf conductance of the whole canopy. The gc is normally estimated from leaf-based measurements of stomatal conductance (gs) using hand-held porometers or gas exchange instruments to be then scaled to the whole canopy using the leaf area index (LAI) [13]. Unfortunately, both measurements (Ψs and gs) are time consuming, require specialized instrumentation and trained personnel, allowing few spot measurements from sentinel plants. The latter characteristics make it difficult to use them for continuous measurements or automated monitoring of plant water stress to obtain a high temporal and spatial resolution and to account for the natural variability of orchards [13,14].

During the last three decades, advances in technology and low-cost portable devices such as thermal infrared (TIR) cameras have been used as effective tools to estimate leaf temperature (and the canopy temperature, Tc), which has been recognized as a reliable proxy for gs and an indicator of plant water status [13,15,16,17,18,19,20]. The latter concept is based on the knowledge that leaf temperature cools down with a higher transpiration rate and gs and, conversely, warms up at lower gs. However, these parameters cannot be estimated directly with a TIR device without previously separating non-leaf material, such as branches, sky, or soil from the analysis [18,21]. To avoid this technical inconvenience, based on the work of Idso et al. [22], other physiological indicators such as the Crop Water Stress Index (CWSI) has been developed based on the extraction of high-temperature, or “dry leaf” (T_dry_), and low- temperature, or “wet leaf” (T_wet_) thresholds representative of the specific canopy’s physical properties. To extract T_dry_ and T_wet_, mature and fully expanded leaves are physically “painted” with water and petroleum jelly, respectively, to obtain these thresholds manually from the TIR images [13,23]. These thresholds allow normalizing the CWSI, resulting in values between 0 and 1, which represents non-stressed and stressed conditions, respectively.

TIR technology corresponds to a remote sensing and a versatile tool to investigate abiotic plant stress [24]. In general, TIR devices have a sensitivity in the mid- and long-wave light spectra from 3000–14,000 nm [25], which allow the scan of surface temperatures within every pixel from the whole TIR image. However, high-resolution TIR cameras are still considered expensive, with price differences of over 80% relative to low-resolution devices, which makes it difficult to implement them for the practical monitoring of plant water status in agriculture. The low-resolution TIR cameras have been evaluated as an alternative for plant water status detection and estimation with promising results for grapevines [16,26,27,28] and almond trees [15,29,30]. New, low-cost, portable mini TIR cameras that come built-in or as accessories of smartphones are now available at affordable prices, offering the opportunity to develop and apply image-based algorithms to improve the application of RDI in cherry trees.

Nowadays, the majority of fruit growers have smartphones, which have these low-cost TIR cameras or can add them. This technology offers a unique opportunity to make the measurement/management of the water status of cherry trees a straightforward process. Thus, the main aim of this study was to compare traditional water status indicators (Ψs and gs) against Tc, and the CWSI computed from TIR cameras of low and high resolution during the post-harvest period in adult cherry trees. A secondary aim was to evaluate whether differences in vegetative habit and vigor that characterize both cherry cultivars studied influence the CWSI’s performance in comparison to more traditional physiological water status indicators, such as Ψs and gs. It is hypothesized that either CWSI calculated from a high- or low-resolution TIR camera could be a reliable alternative to replace Ψs and gs in describing the water status behavior of adult cherry trees.

## 2. Materials and Methods

### 2.1. General Description

The study was carried out during the 2017–2018 and 2018–2019 growing seasons (September to March in the Southern Hemisphere). The climate in the zone has been classified as semiarid, warm, suprathermal with a thermal regime that has a maximum of 29.7 °C and minimum temperatures of 5.2 °C, in January and July, respectively. On average, there are at least five frosts events yearly, with a frost-free period starting from mid-October of about 234 consecutive days. The average annual precipitation is 645 mm on average, mainly concentrated in the fall–winter season. The dry period is from spring to summer, with almost null rain events, resulting in a water deficit of 1027 mm year^−1^ [31].

The experimental plot was located inside a cherry trees orchard at Comalle, Curicó, Chile (34.82° LS; 71.29° LW; 325 m. a. s. l.). The soil is classified as Comalle series (fine, mixed, thermic Aquultic Haploxeralfs) [32]. Soil samples were collected from pits dug in the tree row space to obtain the volumetric soil water content (θ) at field capacity (θFC) and wilting point (θWP), which represents the root zone (≈1.00 m depth). From measurements of soil texture and organic matter analysis, the estimations were calculated as θFC = 40.30% and θWP = 28.03% using the Spaw hydrology software [33].

The entire field has a total surface of 13.2 hectares (ha), from which 2.8 ha are planted with the cultivar ‘Regina’ and 2.2 ha with ‘Sweetheart’, both grafted on ‘Sour cherries’ (*Prunus cereasus* L.) rootstock. The cultivar ‘Regina’ is characterized by presenting vigorous trees, with very long stems, whereas ‘Sweetheart’ presents a medium vigor, where the tree habit spreads with few branches [1]. The trees were planted in rows oriented from north to south in 2008 (9-year-old trees at the beginning of the experiment). The plant density is 741 plants ha^−1^ (3.0 m between trees and 4.5 m between rows). Tree heights are approximately 3.2 m on average, and the canopy is trained on Solaxe system. The Leaf Area Index (LAI) in the period of study was sampled before the end of each growing season (at the beginning of fall) from six trees randomly selected at each cultivar, and estimated using the cover photography method proposed for cherry trees by Carrasco-Benavides et al. [34]. The resulting LAI was 2.23 ± 0.55 and 2.58 ± 0.72 m^2^ m^−2^ for ‘Regina’ and ‘Sweetheart’.

Each tree was drip-irrigated with 4 L h^−1^ drippers separated every 0.5 m and using two drip lines per tree spaced at 0.5 m with the tree trunk in the middle. The irrigation cycle for the whole cherry tree orchard was performed once or twice a week, according to the farm irrigation staff recommendations. The decisions of irrigation frequencies and volumes were based on daily reference evapotranspiration (ETo) data, combined with soil moisture samples. Daily ETo values were provided from an automated weather station (AWS) (VANTAGE PRO2, Davis Instruments, Hayward, CA, USA), placed at approximately 1.9 km apart from the experimental plot. It is important to emphasize that irrigation scheduling, the vigor of trees, fertilization, diseases, and pest management, and all other field labors were not part of this study and followed the standards of the farm management.

### 2.2. Experimental Trial and Field Measurements

An irrigation–restrictions cycle (IRC) experiment was imposed at each studied growing season, from the post-harvest (beginning of January) until leaf fall (end of March). The IRC experiment design consisted in a Randomised Complete Block Design with three treatments and three blocks: T0 = control (Ψs ≥ −1.0 MPa); T1 = low–medium water stress (−1.0 ≥ Ψs ≥ −1.5 MPa) and T2 = high (severe) water stress (−1.5 > Ψs ≥ −2.0 MPa). The selected Ψs thresholds were based on previous RDI studies on stone fruits [3,35]. All experiments were established in a single tree row per each cultivar. Each of the three treatments were imposed on 48 trees divided into four replicates of four trees for each experimental unit. From each experimental unit, only the two central trees were measured (*n* = 24 measured trees per cultivar), keeping the first and final trees as guard rows. The potential effects of lateral infiltration from the adjacent rows were assessed from additional measurements of Ψs, which was carried out weekly at noon from adjacent trees from each treatment (data not shown).

As mentioned previously, irrigation frequency and time were applied according to the standards considered for the whole field, and it was assumed as the T0. Thus, to impose the treatments T1 and T2, irrigation valves were installed in each drip line to open and close them manually until reaching the desired Ψs threshold. After the T1 and T2 cherry trees obtained their respective Ψs thresholds, a recovering water stress cycle was applied, by opening the valves, until the trees reached Ψs similar to T0. Then, a new water restriction cycle started by closing the valves again. This irrigation–restriction dynamic was repeated during each entire post-harvest period until the end of the growing season.

All field measurements were carried out once a week during the previous day of irrigation and performed only on clear days, avoiding days that presented partial and cloudy weather conditions, following recommendations from previous studies [20,36,37]. These measurements were performed between 1100–1500 h (around solar noon) to ensure that the trees were under the maximum atmospheric water demand [12,15]. The Ψs were measured using a pressure chamber (model 600, PMS instruments, Albany, OR, USA) in a mature, fully expanded, and healthy leaf per each tree. Selected leaves were firstly wrapped in plastic and then in aluminum foil for 1–2 h before measurements to equalize leaf water potential (Ψ_L_) with Ψs [10,12,38]. The gs was measured on another two mature, fully expanded, and healthy leaves, one from the sunlit side and the other from the shaded side of the trees. These measurements were carried out by using a portable leaf porometer (SC-1, METER Group, Inc., Pullman, WA, USA). At the same time, TIR images were obtained from the whole tree canopy side (horizontal measurement technique ≈ 3.5 m, Figure 1a), focusing on the center of the canopy. Thermal images were taken using: (1) A high-resolution (HR) camera (TIS60, Fluke Corporation, Everett, WA, USA) with a 260 × 195-pixel resolution and sensitivity of 7.5–14 μm spectral range, and, (2) a low-resolution (LR) camera (Lepton module, FLIR Systems Inc., Wilsonville, OR, USA) built into a smartphone (CAT S60, Catphones Mobile Ltd., Reading, Berkshire, UK), with an 80 × 60-pixel resolution and sensitivity of 8.0–14 μm spectral range. For both cameras, the emissivity (ε) value was set at 0.95, similar to García-Tejero et al. [30]. All physiological measurements were obtained within 2 to 4 min after the TIR image shot, following the suggestions of Jones [39], which showed that during this time, it is unlikely that gs will significantly change unless there were winds of 5 m s^−1^.

To record the seasonal variations of field meteorological conditions, an Automatic Weather Station (AWS) was installed inside the experimental plot. The sensors and height for measure each variable were configured as follows: two air temperature and relative humidity (Ta, RH) probes (VP-4, METER Group, Inc., Pullman, WA, USA), one placed inside the canopy (at 1.5 m), and the other at 1.0 m above the top of tree canopies, respectively; a wind speed (u), and wind direction (wd), sensor (Wind, Davis Instruments, Hayward, CA, USA) placed at 2 m height in the inter-row space; a sensor of precipitation (Pp) (ECRN-50, METER Group, Inc., Pullman, WA, USA), placed at 1.5 m height in the inter-row space, and a silicon pyranometer sensor (solar radiation -Rs) (PYR, METER Group, Inc., Pullman, WA, USA), placed at 1.5 m above the top of the canopy. All meteorological data were half-hour averaged and stored in a data logger (Em 50, METER Group, Inc., Pullman, WA, USA).

### 2.3. Thermal Images Post-Processing and CWSI Computations

Thermal images of the tree canopies typically include pixels with non-leaf material such as branches, stems, trunk, soil, and sky, which need to be extracted by masking them out using computer vision algorithms [13]. The baselines to compute CWSI was obtained by an empirical approach, where T_wet_ and T_dry_ thresholds were obtained by painting two representative healthy and non-detached mature leaves on both sides (adaxial and abaxial) with water mixed with dishwashing soap (0.01% v/v) and liquid petroleum jelly, respectively [13] (Figure 1), two minutes before the TIR image acquisition. From each TIR image, the CWSI was computed as follows:(1)$$CWSI=\frac{\left(T_{c}-T_{wet}\right)}{\left(T_{dry}-T_{wet}\right)}$$
where T_wet_ and T_dry_ are the wet and dry temperatures captured by the thermal image, respectively (all in °C). If CWSI = 0, it means that T_c_ = T_wet_ indicating that the plant is fully transpiring without water stress. Conversely, when CWSI = 1, the plant is dry because T_c_ = T_dry,_ which means that stomata are partially or fully closed.

Thermal images obtained from the HR and LR TIR cameras were exported as Comma Separated Values (csv) file using the SmartView (version 4.3.79.0, Fluke Corporation, Everett, WA, USA) and FLIR Tools (version 6.4.17317.1002, FLIR Systems Inc., Wilsonville, OR, USA) software, respectively. Then from each *.csv file, each TIR image was reconstructed by using a customized script developed in Matlab© R2019a (Mathworks Inc., Natick, MA, USA). This script allowed the handy selection of the dry and wet leaves in each image, computing the CWSI (using Equation (1)). All the results were exported as another *.csv file and a grayscale Joint Photographic Experts Group (JPEG) image for the posterior analysis.

### 2.4. Statistical Analysis

In order to describe the relations among raw data of Ψs, gs, and Tc (for both high and low resolution), Pearson’s correlation index (r) was applied. No outlier’s treatment of values was used in the post-processing of the raw data, to show the actual performance of them. Next, the analysis of variance (ANOVA) was applied to determine whether there were any statistically significant differences between the treatments, considering the daily trends of each physiological indicator. When the ANOVA indicated significant differences among treatments, the Tukey’s honest significant difference (HSD) test was carried out at 95% significance. Finally, the comparisons between Tc_HR_ vs. Tc_LR_ and CWSI_HR_ vs. CWSI_LR_ were computed by the Deming’s linear regression method [40]. This method analyzed if the relationship between the two devices was linear (two-tailed test). It also comprised the comparisons of the model coefficients, intercept (a) and slope (b), between data from both devices (confidence interval 95%). This last analysis allowed to check whether the values of Tc and CWSI obtained from the different resolution TIR cameras gave results that are similar or not. Deming’s method allows detecting the systematic differences between the two thermal cameras, considering that both have measurement errors. All analyses and Figures were performed using the RStudio software (RStudio Inc., Boston, MA, USA), and the package ggplot2 [41].

## 3. Results

### 3.1. Hourly Evolution of Weather Conditions

During the post-harvest experiment, for both growing seasons studied, the weather was classified as dry and hot (Figure 2). The hourly based progression of Rs, Ta, RH, u and VPD, evidenced that, during almost all evaluated days, the cherry trees were under clear sky days conditions, presenting less than five cloudy days for each evaluated period (Rs < 500 W m*^−^*^2^). At the time of the tree’s measurements (highlighted in the segmented red boxes), values of Rs ranged between 800–1100 W m*^−^*^2^ were observed (Figure 2a,b). The trends in the values of Ta and RH were both skewed to the right, presenting the highest of Ta by above 30 °C close the 1700 h (Figure 2c,d). The average RH, at the time of field measurements, was close to 50% (Figure 2e,f). The wind speed at the time of measurements was similar between both growing seasons (Figure 2g,h), showing hourly averaged values beneath 1 m s*^−^*^1^ for both growing seasons. The hourly oscillation of the VPD reached values by near 3.0 kPa (Figure 2i,j) during the time of measurements, presenting daily peaks of above 4.0 kPa between 1600 and 1800 h for both post-harvest growing seasons.

### 3.2. Relationships among Raw Data

Figure 3 and Figure 4 depict the relationships among the noontime averages of Ψs, gs, and Tc, for ‘Regina’ and ‘Sweetheart’ cherry trees, during both evaluated growing seasons. In the case of the ‘Regina’ cherry trees, they presented a significant positive relation between Ψs and gs for all treatments (0.38 < r < 0.70), considering both measurements of gs, from sunlit and shaded leaves, during the first (Figure 3a,b) and second growing season (Figure 3i,j). As expected, the lowest values of Ψs and gs were observed in T2, where individual trees reached a minimum Ψs of −2.6 MPa, presenting the lowest range of gs between 131 and 175 mmol m^−2^ s^−1^, for shaded and sunlit leaves. Likewise, the relation between Ψs and Tc, computed from high- and low-resolution thermal cameras, showed significant negative correlations, with an average r of −0.56 ± 0.04 (Tc_HR_) and −0.50 ± 0.03 (Tc_LR_) (Figure 3c,d, and Figure 3k,l, for the 2017–2018 and 2018–2019 growing seasons). In general, the values of Tc_LR_ were lower than Tc_HR_, ranging from 19 to 32 °C, in comparison to that presented in Tc_HR_, which were between 26 and 39 °C for both growing seasons. For the raw comparison between gs from sunlit leaves and both Tc (Figure 3e,g (2017–2018), and Figure 3m,n,o (2018–2019)), it can be observed that there were significant negative relations, both for high- and low-resolution TIR cameras (r ≈ −0.41 ± 0.03). In contrast to the sunlit leaves, for the shaded ones, these correlations were also significant, but lower (r ≈ −0.36 ± 0.06) (Figure 3f,h,n,p).

In general, the ‘Sweetheart’ cherry trees presented flatter patterns compared to the ‘Regina’ trees. Considering the water stress–recovery cycles, ‘Sweetheart’ exhibited values of Ψs in the range of −1.0 to −2.0 MPa, and gs between 250–1000 mmol m^−2^ s^−1^ (Figure 4). The correlations between Ψs vs. gs were low but significant (r ≈ 0.39–0.56) always for the measurements from sunlit leaves (Figure 4a,i). On the contrary, this correlation was only significant (r = 0.38) for the shaded leaves in the first growing season (Figure 4b). The analysis of the relations between Ψs and Tc, from both high- and low-resolution thermal cameras (Figure 4c,d,k,l), presented significant but low correlations than those observed for the Ψs against gs, resulting in Tc_HR_: r = −0.30 ± 0.23; Tc_LR_: r = −0.35 ± 0.07. When comparing the correlations between Tc and gs, they were not always significant. In this regard, the best results for the gs from shaded leaves, compared to Tc_HR_ and Tc_LR_, were observed during 2017–2018 (−0.23 < r < −0.21) (Figure 4f,h). On the contrary, the best correlations were observed for the sunlit leaves, comparing both TIR cameras for the second growing season (−0.32 < r < −0.29) (Figure 4m,o). Although these relations were significant, the results presented herein were lower than the same observed for ‘Regina’ cherry trees.

### 3.3. Seasonal Behavior of Physiological Indicators: Stress–Recovery Cycles

The performance of different physiological indicators used to describe the post-harvest dynamics of water stress–recovery cycles is shown in Figure 5 and Figure 6. In general, at noon, the behavior of Ψs and gs, showed similar trends for both cultivars during the complete study. The dynamics of gs followed similar trends compared to Ψs, especially in the measurements from sunlit leaves (Figure 5c,d). However, averaged gs were less sensitive to detect significant differences among treatments. The trends of Ψs and gs, for ‘Regina’ cherry trees (Figure 5a–f) showed significant differences of Ψs for T0 vs. T2, but it was not the case for T1, showing that the trees of T1 and T2 were under similar water stress levels during some periods in the post-harvest season (Figure 5a,b). Despite the use of control valves to manage the stress-irrigation cycle in T1 and T2, there were days with extreme weather conditions (daily Ta maxima > 30 °C, RH minima < 20%), that made difficult to control the middle-level treatment (T1), without getting close to the most restricted treatment (T2) (e.g., day of the year (DOY) 40 = Ψs < −2.0 MPa; gs ≈ 300 mmol m^−2^ s^−1^). On the other hand, the seasonal behavior of Ψs and gs for ‘Sweetheart’ cherry trees showed significant differences among the imposed treatments for almost all DOY (Figure 6a–f). During both growing seasons, the averages of Ψs for T0 differed from T1 and T2. Similarly to ‘Regina’ cherry trees, no differences were observed in the comparisons between T1 and T2 (Figure 6a,b). In the same way, measurements of gs at noon, described similar patterns as Ψs, presenting fewer days with significant differences than this parameter. These results may show that gs values, especially for those measured from sunlit leaves, were better than the shaded ones, to highlight the differences among treatments (Figure 6c,d). The results presented in Figure 6, evidenced that in ‘Sweetheart’ trees, the aimed thresholds of Ψs were not reached during the entire experiment. During the whole post-harvest period, T2 never reached the desired limit (−1.5 > Ψs ≥ −2.0 MPa), neither in the 2017–2018 nor in the 2018–2019 growing season, despite the control valves were closed all the time.

### 3.4. Comparisons Between High- and Low-Resolution Thermal Images

The comparisons between the Tc and CWSI computed from high- and low-resolution TIR cameras showed that in both cultivars (‘Regina’, Figure 7a–d, and ‘Sweetheart’, Figure 7e–h), the values of Tc_HR_ vs. Tc_LR_ were beneath the 1:1 line. In this regard, in almost all comparisons, there were linear relations, except between Tc_HR_ vs. Tc_LR_ for both cultivars during 2018–2019 (Table 1). Additionally, no linear relationships were observed for the CWSI’s calculated for ‘Regina’ during the same period. The analysis of the slope showed that on average Tc_HR_ was higher in around 16% than Tc_LR_ (Figure 7a,c,e,g), demonstrating that it is unlikely that both cameras yield the same results when comparing the values of Tc. These differences between the values of Tc influenced the CWSI’s, which were also higher for those calculated from Tc_HR_ images (Figure 7b,d,f,h). These plots show that the distribution of dots for the comparison of both CWSI’s trended to be spreader than Tc’s, and slightly below 1:1 line, evidencing higher values for CWSI_HR_ in contrast to CWSI_LR_. The analysis of the slope of CWSI_HR_ to CWSI_LR_ presented in Table 1, revealed that, for both cherry tree cultivars, the resulting CWSI_HR_ were, on average, 11 ± 0.27% higher than CWSI_LR_.

## 4. Discussion

The stability of daily weather conditions is critical when estimating the water status of plants using TIR devices. The quality of the CWSI estimations is more accurate when the TIR measurements are done under clear sky, dry and not windy conditions [13,15,20,27,29,37,39,42]. It is well known that when the wind speed increases, the accuracy of CWSI decreases due to the rapid changes in gs, which affects the leaves’ transpiration, increasing leaf temperature [39,43]. In addition, when weather conditions present low Ta and high RH, resulting in low VPD, the performance of CWSI could be questionable [20,39]. Considering the high dependence of this technique of the weather condition, it should be highlighted that during this experiment, the weather conditions in all the evaluated days at the noontime were optimally presenting sunny, dry, hot, and non-windy days (Figure 2).

Ψs and gs have been recommended as integrative tools of the soil–plant–atmosphere continuum due to their quick response to the water availability in the soil [44,45,46]. In the same way, TIR technology has been recommended as another indicator of the tree’s water status, because the more water stress implies higher stomatal closure, increasing the leaf’s temperature [20,43,47]. This theoretical background agrees with the depicted linear relations for raw data of Ψs, gs (sunlit and shaded leaves), and Tc (low and high resolution) presented in Figure 3 and Figure 4. As the water stress went forward, the Ψs and gs decreased, increasing Tc. These cause–effect relationships were significant in all indicators for ‘Regina’ and almost all in ‘Sweetheart’. Pearson’s correlation indexes between Ψs vs. Tc were in the same range to other studies in woody crops such as almond (−0.90 < r < −0.66, [29,30], and olive trees (r = −0.73, [45]). The relations between Tc and gs were also negative and in the range of the observed in grapevines and olive trees (−0.90 < r < −0.71, [45,48]). These correlations values vary from study to study because the TIR images caption and post-processing are done in different ways. Depending on the distance from the canopy, each image can include woody elements that add noise to the averages estimations of Tc and CWSI. Some studies prefer to use overall averages to smooth the natural dispersion of points, which may improve the results among the different indexes. As there is not an unique way to address the image process issue, in this study, the existence of significant relations was considered enough to describe the relationships among the different physiological indicators.

The dynamics of water–stress recovery cycles of cherry trees during the post-harvest period showed similar patterns for Ψs, gs, and inversely for both high- and low-resolution CWSI’s (Figure 5 and Figure 6). Theoretically, at each water stress–recovery cycle, when irrigating, there is no water stress, Ψs and gs would be high, and CWSI would show an inverse pattern. Then, when the trees were under water stress, each physiological indicator would show the opposite. This behavior was observed when comparing the seasonal trendings of Ψs, gs, and CWSI’s as well as for ‘Regina’ and ‘Sweetheart’. Specifically for ‘Regina’ cherry trees, both CWSI’s, computed from HR and LR thermal cameras (Figure 5g–j), imitated the general behavior of Ψs and gs, reaching values of CWSI between 0.60 and 0.75, for the most stressed trees (T2) in comparison to T0 (CWSI’s < 0.50). The ‘Regina’ trees were the most affected by water stress, which allowed a detailed analysis of CWSI performance compared to other physiological indicators. Figure 5 shows that both high- and low-resolution CWSI’s presented fewer days with significant differences in the water status of trees, similar to those from Ψs and gs (Figure 5a–f). Indeed, less than three days showed significant differences for CWSI’s, especially for CWSI_LR_ (Figure 5i,j), showing that CWSI was not robust enough to detect differences in treatments, in comparison to the other physiological parameters. For ‘Sweetheart’, results showed flat patterns in almost all physiological indicators (Figure 6). This behavior showed that the trees on the ‘Sweetheart’ experiment did not have substantial water stress exposure, to achieve significant and measurable changes. A possible explanation of the latter could be the presence of an intermittent water table in the area where the ‘Sweetheart’ experiment was performed, which was outside of the control of this experiment.

The global performance of the CWSI’s of this study was in accordance to those obtained for olive trees [21,49], table grapes [37], pistachio trees [20], and almond trees [15]. Hence, the behavior of CWSI obtained was as expected, due to different water regimes, the comparison of means only presented significant differences in some days, in contrast to Ψs, gs (Figure 5 and Figure 6). The CWSI varies depending on changes in leaf surface temperatures obtained from the TIR images, which is more precise when the water stress is medium to high [21,50]. Particular aspects of CWSI that could be related to the water status of pistachio trees have been highlighted by Testi et al. [20]. They observed that this indicator was not entirely stable day-to-day and slightly dropped when the non-water stressed trees were watered. Contrarily, this behavior changed rapidly in the most stressed treatment, where the recovery, although fast, was incomplete, presumably by temporary xylem cavitation. For this study, the magnitude of the recovery shown by the CWSI, in particular for ‘Regina’, was evident in comparison to the curves of Ψs, independent of the camera resolution (Figure 5).

In thermography, it is recognized that the higher number of pixels, provides more information for an accurate thermal interpretation of the scene [25]. According to the manufacturers, the low-resolution thermal camera used in this experiment offers an image size that is 90.5% (4800 pixels) smaller than the high-resolution one (50,700 pixels). The high-resolution TIR camera has a thermal sensitivity of ≤0.08 °C, an accuracy of ±2 °C, and a temperature measurement range of −20 to 550 °C. Similarly, the low-resolution thermal camera offers a thermal sensitivity <0.05° C, an accuracy of ±3 °C, and a thermal range of 0 to 120 °C. These differences between both devices did not significantly affect the daily performance of the CWSI computation, which could follow the dynamics of the post-harvest water status of cherry trees as an inverse proxy of the behavior of Ψs and gs (Figure 5 and Figure 6). Each low- and high-resolution TIR image was captured at the same distance and angle, trying to capture all elements of the center of the canopy. Despite that, both CWSI’s were not sensitive enough to slight variations among treatments in the same way as Ψs, and gs, detecting fewer days with significant differences. The comparison of thermal images of different resolution applied to monitor the water status of woody crops has been recently explored with contrasting results. For example, Fuentes et al. [26] obtained coefficients of determination (r^2^) > 0.91 for CWSI computed in grapevines and olives trees by contrasting high- and low-resolution TIR images. When the same contrast was made against an IR scanner (10 × 10 pixels), the performance of CWSI decreased, resulting in r^2^ = 0.68, for olive trees, and r^2^ = 0.37 for grapevines. In almond García-Tejero et al. [30] obtained r^2^ = 0.90 when compared averages of computed from two different resolution TIR cameras. In the study by García-Tejero et al. [30], there are no results about the CWSI’s analogy. These studies applied the traditional simple linear regression, which assumes that only the response variable was measured with error, which could be not entirely true when comparing two methods to measure the same variable [40]. In fact, the linear regression method may be used for calibrating a new technique against an established one, but it is not recommended when comparing thermal cameras of different resolutions. The linear regression method is recommended when a new method is compared to a ground truth and assumes that the sources of error are from the new method [51]. Hence, when comparing two thermal cameras, both have sources of error; then, the Deming regression method is better suited for comparison purposes. Other thermal imagery comparisons obtained from two different resolution TIR cameras for biomedical research [52] showed that low-cost smartphone-based thermal cameras could be an excellent alternative to high-resolution ones, only after proper calibration. This previous calibration was not considered in our study because, under actual field conditions, it is almost impossible for a fruit grower use this method. The factory calibration of both TIR cameras was used, under the assumption that the normalization procedure, embedded in the CWSI calculation using the dry and hot leaves per image, could reduce the biases.

## 5. Conclusions

The results presented in this study showed that low-resolution thermal cameras can be used as a suitable, accurate, and affordable alternative to monitor the water status of cherry trees. Neither the differences in the resolution of thermal cameras nor the cultivar influenced the post-harvest CWSI performance. Thus, the CWSI computed from low-resolution TIR cameras could be suggested as a practical, affordable, fast, and accurate tool to assess the water status trends through the growing season of ‘Regina’ and ‘Sweetheart’ cherry trees in the field for precise irrigation management purposes.

## Figures and Tables

**Figure 1 sensors-20-03596-f001:**
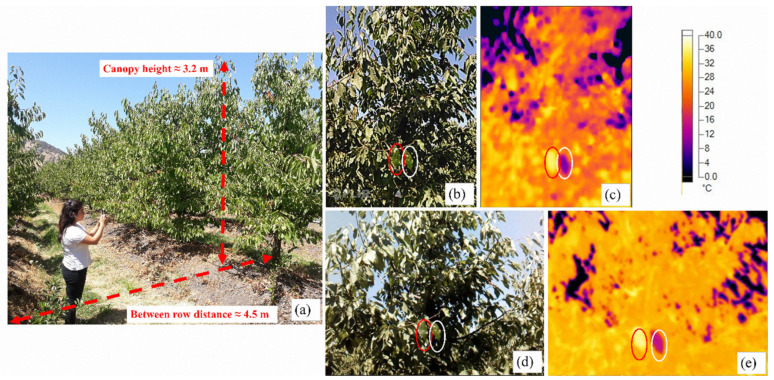
Examples of (**a**) image shot from the side of each tree row; false-color thermal images obtained from low (**c**) and high resolution (**e**) thermal cameras, and the corresponding RGB images (**b**,**d**). The “wet leaf” is depicted in each thermal image using a white circle; meanwhile, the “dry leaf” is shown in a red circle. Both images (**b**–**e**), are from the same tree, and they were captured at the same time using the two thermal cameras, the day of the year (DOY) 44, in 2018.

**Figure 2 sensors-20-03596-f002:**
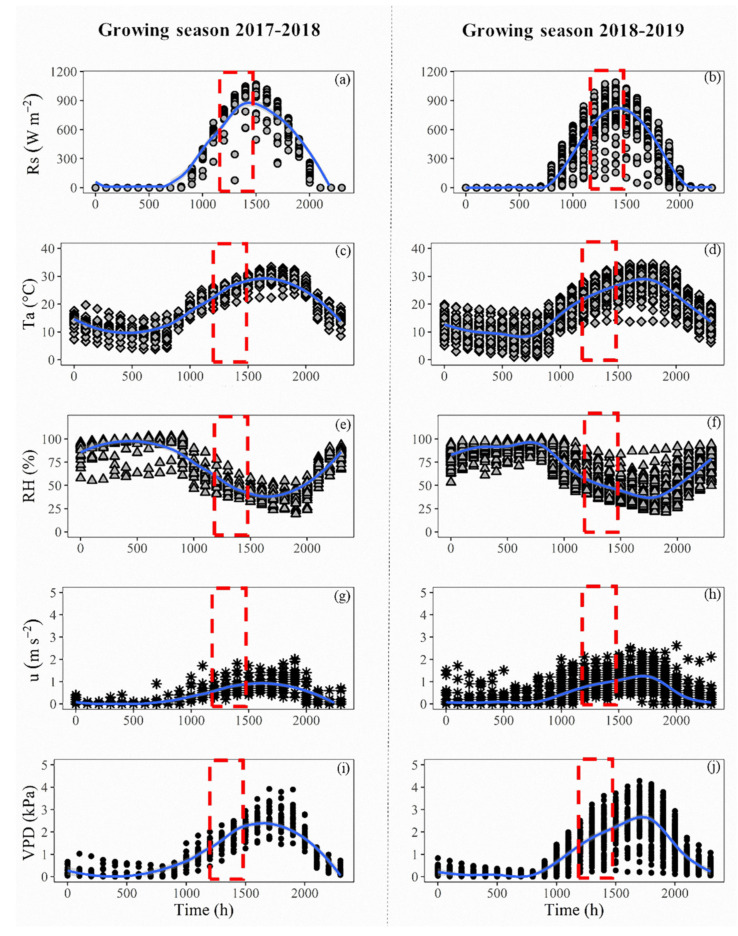
Post-harvest hourly variations of (**a**,**b**) solar radiation (Rs), (**c**,**d**) air temperature (Ta), (**e**,**f**) relative humidity (RH), (**g**,**h**) wind speed (u), and (**i**,**j**) vapor pressure deficit (VPD), inside the studied cherry trees orchard (growing seasons 2017–2018: left side, and 2018–2019: right side). The segmented red boxes denote the time of plant field measurements. Blue lines indicate the average hourly value.

**Figure 3 sensors-20-03596-f003:**
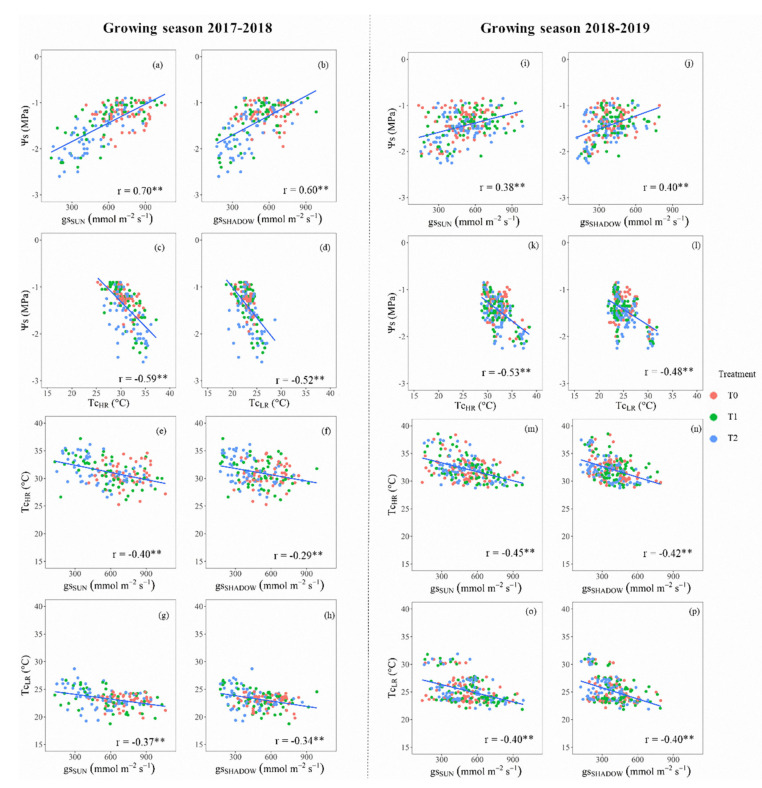
Linear relationships among average values of midday stem water potential (Ψs), stomatal conductance (gs) for sunlit (SUN) and shaded (SHADOW) leaves, and canopy temperature (Tc) for cherry trees ‘Regina’ for each post-harvest irrigation treatment (control: T0 (Ψs ≥ −1.0 MPa); low–medium water stress: T1 (−1.0 ≥ Ψs ≥ −1.5 MPa), and high (severe) water stress: T2 (−1.5 ≥ Ψs ≥ −2.0 MPa). Subscripts HR and LR, denote that the thermal images were shot from a high- and low-resolution TIR camera. Growing seasons 2017–2018: left side, and 2018–2019: right side. Levels of significance: 0.01 **. (**a**,**i**) Ψs vs. gs_SUN_, (**b**,**j**) Ψs vs. gs_SHADOW_, (**c**,**k**) Ψs vs. Tc_HR_, (**d**,**l**) Ψs vs. Tc_LR_, (**e**,**m**) Tc_HR_ vs. gs_SUN_, (**f**,**n**) Tc_HR_ vs. gs_SHADOW_, (**g**,**o**) Tc_LR_ vs. gs_SUN_, and (**h**,**p**) Tc_LR_ vs. gs_SHADOW_

**Figure 4 sensors-20-03596-f004:**
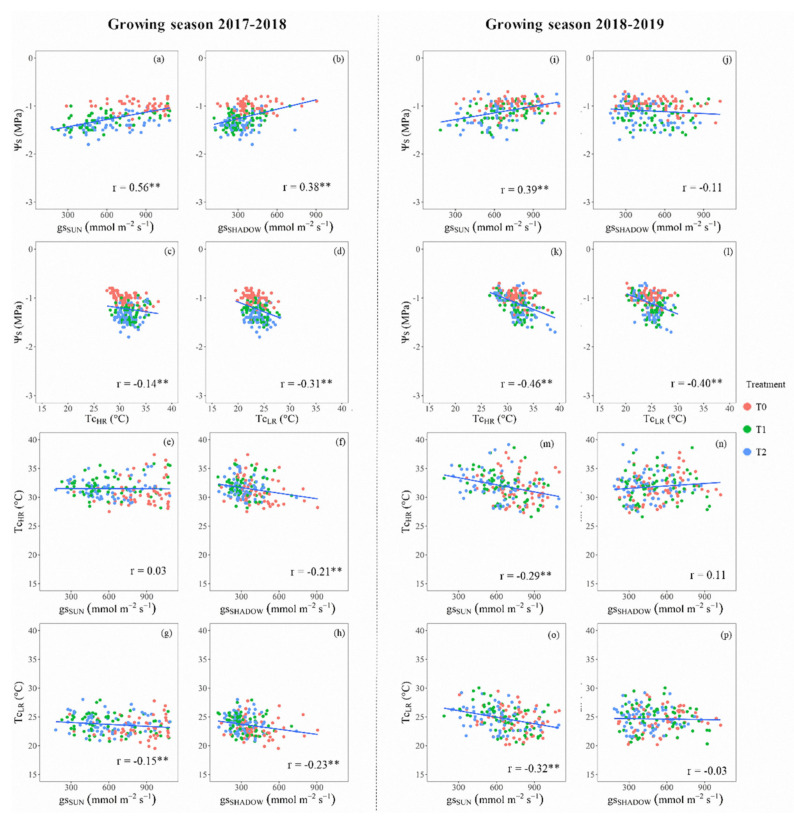
Linear relationships among average values of midday stem water potential (Ψs), stomatal conductance (gs) for sunlit (SUN) and shaded (SHADOW) leaves, and canopy temperature (Tc) for cherry trees ‘Sweetheart’, for each post-harvest irrigation treatment (control: T0 (Ψs ≥ −1.0 MPa); low–medium water stress: T1 (−1.0 ≥ Ψs ≥ −1.5 MPa), and high (severe) water stress: T2 (−1.5 ≥ Ψs ≥ −2.0 MPa). Subscripts HR and LR, denote that the thermal images were shot from a high- and low-resolution TIR camera. Growing seasons 2017–2018: left side, and 2018–2019: right side. Levels of significance: 0.01 **. (**a**,**i**) Ψs vs. gs_SUN_, (**b**,**j**) Ψs vs. gs_SHADOW_, (**c**,**k**) Ψs vs. Tc_HR_, (**d**,**l**) Ψs vs. Tc_LR_, (**e**,**m**) Tc_HR_ vs. gs_SUN_, (**f**,**n**) Tc_HR_ vs. gs_SHADOW_, (**g**,**o**) Tc_LR_ vs. gs_SUN_, and (**h**,**p**) Tc_LR_ vs. gs_SHADOW_

**Figure 5 sensors-20-03596-f005:**
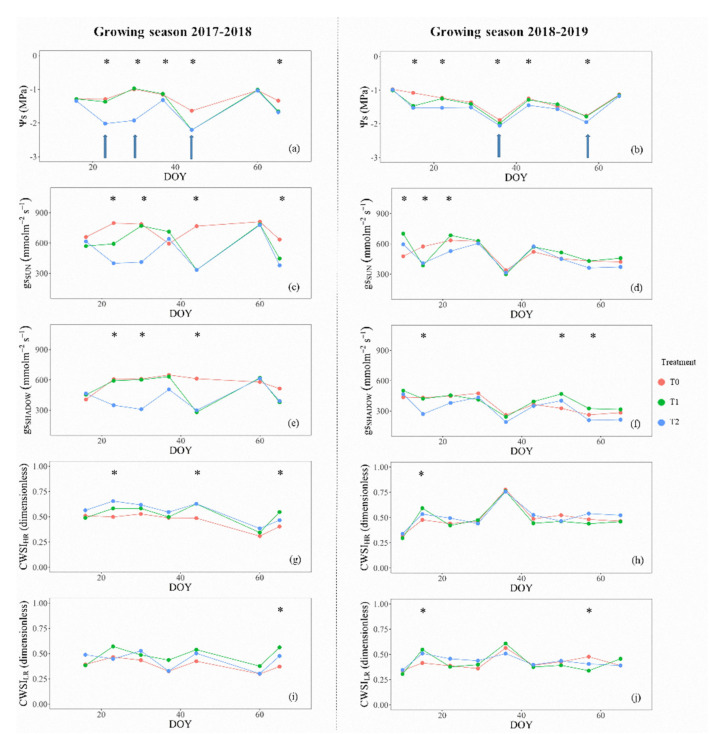
Seasonal evolution of averaged values of midday stem water potential, stomatal conductance, and crop water stress index of cherry trees ‘Regina’, for each post-harvest irrigation treatment (control: T0 (Ψs ≥ −1.0 MPa); low–medium water stress: T1 (−1.0 ≥ Ψs ≥ −1.5 MPa), and high (severe) water stress: T2 (−1.5 ≥ Ψs ≥ −2.0 MPa). (Growing seasons 2017–2018 = (**a**,**c**,**e**,**g**,**i**); Growing season 2018–2019 = (**b**,**d**,**f**,**h**,**j**). Subscripts HR and LR, denote that the thermal images were shot from a high- and low-resolution TIR camera. (* Significant Tukey’s HSD test *p* < 0.05). The arrows indicate the irrigation events for T2.

**Figure 6 sensors-20-03596-f006:**
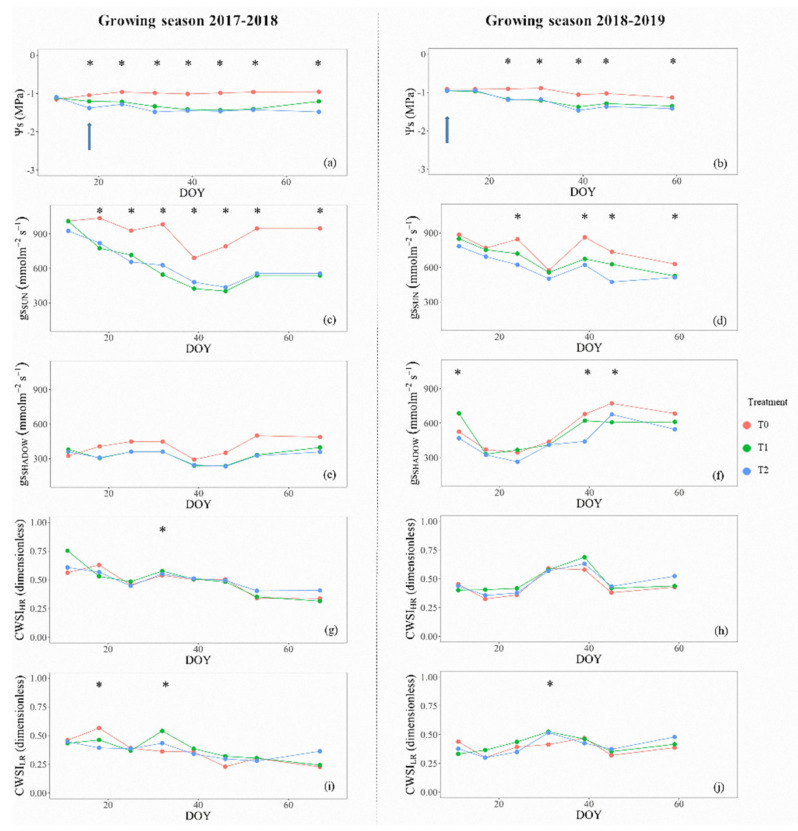
Seasonal evolution of averaged values of midday stem water potential, stomatal conductance, and crop water stress index of cherry trees ‘Sweetheart’, for each post-harvest irrigation treatment (control: T0 (Ψs ≥ −1.0 MPa); low–medium water stress: T1 (−1.0 ≥ Ψs ≥ −1.5 MPa), and high (severe) water stress: T2 (−1.5 ≥ Ψs ≥ −2.0 MPa). (Growing seasons 2017–2018 = (**a**,**c**,**e**,**g**,**i**); Growing season 2018–2019 = (**b**,**d**,**f**,**h**,**j**). Subscripts HR and LR, denote that the thermal images were shot from a high- and low-resolution TIR camera. (* Significant Tukey’s HSD test *p* < 0.05). The arrows indicate the irrigation events for T2.

**Figure 7 sensors-20-03596-f007:**
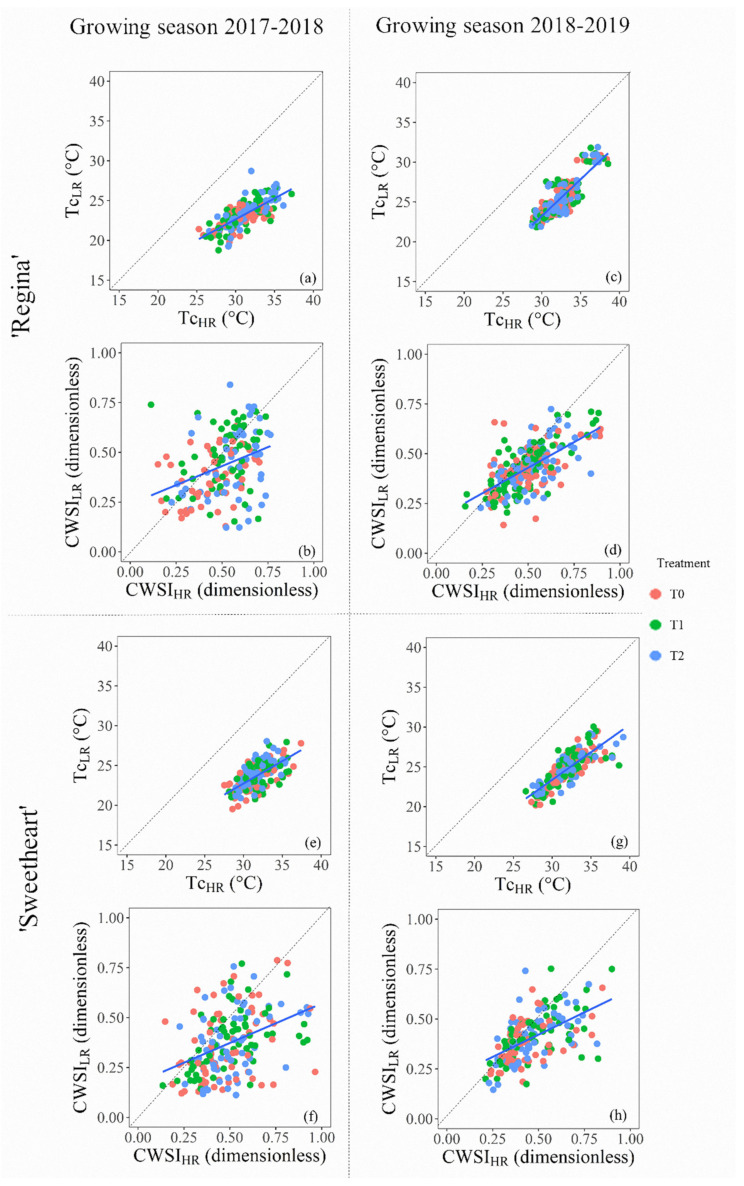
Comparison between data obtained from thermal cameras at high (HR) and low (LR) resolution for ‘Regina’ (superior: (**a**–**d**)) and ‘Sweetheart’ (inferior: (**e**–**h**)) cherry trees under three post-harvest irrigation treatments (control: T0 (Ψs ≥ −1.0 MPa); low–medium water stress: T1 (−1.0 ≥ Ψs ≥ −1.5 MPa), and high (severe) water stress: T2 (−1.5 ≥ Ψs ≥ −2.0 MPa). Segmented lines at each figure depict the 1:1 line. Growing seasons 2017–2018 (left: (**a**,**b**,**e**,**f**)) and 2018–2019 (right: (**c**,**d**,**g**,**h**)).

**Table 1 sensors-20-03596-t001:** Statistical analysis of the comparisons between all data of canopy temperature (Tc) and crop water stress index (CWSI), computed from thermal images of high (HR) vs. low (LR) resolution, for ‘Regina’ and ‘Sweetheart’ cherry trees considering all irrigation treatments.

Cultivar	Growing Season	Variable	Test of Linearity *	Intercept (a) **	*t*-Test	Slope (b) **	*t*-Test
‘Regina’	2017–2018	Tc	T	3.30	F	0.64	F
	2018–2019		F	−10.26	F	1.10	F
	2017–2018	CWSI	T	−0.21	T	1.26	F
	2018–2019		T	0.07	T	0.72	F
‘Sweetheart’	2017–2018	Tc	T	−1.84	T	0.81	F
	2018–2019		F	−1.44	T	0.82	F
	2017–2018	CWSI	T	−0.08	T	0.91	F
	2018–2019		F	0.10	F	0.67	F

* Test of linearity (Two-tailed test): null hypothesis (the relationship between the two variables is linear) True (T); Alternative hypothesis (the relationship between the two variables is no linear) False (F). ** Linear regression parameter analysis: null hypothesis (a = 0; b = 1) T; Alternative hypothesis (a ≠ 0; b ≠ 1) F. *, ** Confidence interval: 95%.

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
