# Peer review of "Performance Assessment of Thermal Infrared Cameras of Different Resolutions to Estimate Tree Water Status from Two Cherry Cultivars: An Alternative to Midday Stem Water Potential and Stomatal Conductance"

_sensors, 2020, doi:10.3390/s20123596_

Round 1

Reviewer 1 Report

This paper deals with a relevant topic for modern horticulture and agriculture in dry areas. Thermography can be an useful tool for practice especially if we consider the increasing availability of low cost solutions in the market. The topic is not novel but it can be relevant for readers of the Journal interested in low cost sensors and related performance.

The paper should explain better why authors tested stress-recovery responses and why they studied two genotypes. In fact, the paper could be shortened if only one cv was analyzed which would make it easier to read and more focused.

There are also in some part of the text the use of an excessive number of citations to support a statement (up to 8 references) which is excessive and should be avoided or substituted by a review. Finally the discussion/conclusion should also present some trends/prospects for the use of low cost thermography and major pros and cons.

The paper is well written and structured but needs still some improvements. Please have in consideration the following points and suggestions for improvement:

  • In the abstract indicate the species (Latin name) as well as the two cvs tested;
  • Line …I would write “water use efficiency and water productivity” to be more correct if you consider the physiological background;.
  • Line 64. Instead of water scarcity I suggest “water stress” or mild to severe water stress”;
  • Line 72, what is gL and nL?;
  • Line 113. Better to write " have smartphones that would permit to install low cost IR cameras on them…”;
  • At the end of the discussion authors should also refer why they tested the two cvs;
  • Line 189. Indicate the cost as it makes it more objective and quantitative;
  • Line 221. Quality of the IR images seems quite similar. The high resolution/quality IR image seems similar to the low resolution? Is there any focus problem?
  • Line 267. The figure 2, could have indicated at the top, the season years in order to be more clear to the reader;
  • Line 283. Why using the decimals for the T values?
  • Line 305. In the legend of the Figure 5 and 6 indicate what does it mean “T0, T1 and T2”; The legend should be informative. Also indicate the models of the cameras used; Instead of “sunny” I suggest “sunlit”;
  • Line 342-343. It should be more clear the stress recovery moments in the graphs; maybe using indicative arrows! The same for figure 6;
  • Line 388. Instead of “varies” use “decreases”;
  • Line 397. Write “" increasing leaf temperature";
  • Line 493. Instead of “fine” write “robust”;
  • The paper misses some discussion about the apparent no effect of genotype and why authors have decided to compare this two cvs;

Reviewer 2 Report

The research is Interest to the readers. However, it presents problems of focus in the description of the results and the discussion. The current approach is mostly from an agronomic perspective, however, the important thing about the publication is the comparison of the performance of each sensor, and its possibility of using it in the future. Overcoming the indicated limitations and the written observations, the research will provide an interesting contribution in the use of different sensors for the management of water stress.

Title: The title is interesting. But I think your article doesn't talk much about the "size, cost, and resolution" of the sensor. That being so, it may be appropriate to modify the title. Consider that in the results and discussion section, it is recommended that you a re-focus.

Abstract: The summary is appropriate. In fact, very well written. I just recommend adding the standard deviation in the percentage of variation of your results, and checking the keywords.

Introduction: The first part of the introduction is adequate, since it allows contextualizing the research. But in the end, the introduction took the form of a specific bibliographic review on the operation of certain parameters. I think it more appropriate to consider an analysis of the state of the art of the use of sensors, by taking images of different quality, to define water stress.

Materials and Methods: The section is appropriate. But it is recommended to improve the order in the delivery of information, to facilitate understanding and not cause confusion. In the specific comments, recommendations were left.

Results: The results are interesting. It is recommended to improve the presentation of some figures, in order to facilitate their understanding (see specific comments). Similarly, it is recommended to add a table with the statistical results, to understand the behaviour of the different sensors (see specific comments). The most importantly. I consider that the focus of the results should be focused on the performance of the different sensors used (compared to each other) and their comparison with direct measurements of the parameters. Make distance from the agronomic comparison between varieties.

Discussion: The discussion is deep and interesting, particularly towards the end of the section. I recommend that, like the results, the discussion focuses on the performance of the different sensors used (compared to each other) and their comparison with direct measurements of the parameters. I also recommend reference the corresponding figures and tables, when discussing the results.

Conclusions: I think the conclusion could be improved once you make the focus changes to the results and discussion. There is a contradiction between the first, second and third sentences (check it; see specific comments).

Round 2

Reviewer 2 Report

The changes made in the writing have managed to improve the focus of his research. I appreciate the answers and clarifications.

Author Response

Dear Reviewer 2,

We have received your new comments and minor editing requirements on our manuscript Sensors-821026: “Performance assessment of thermal infrared cameras of different resolutions to estimate tree water status from two cherry cultivars: An alternative to midday stem water potential and stomatal conductance”. We sincerely appreciate your valuable suggestions and your rigorous observations to the modified manuscript. All the modifications have been highlighted in blue in this response letter and in the new version of the manuscript, as follows:

Comments Reviewer 2

Other possible explanations may also be linked to the adaptation

differences that the variety could present. There could be a bibliography about it.

Answer: Indeed, in the introduction section it appears a short bibliographic description of both cultivars of cherry (Please check lines 50-52, and 139-140). We only found those references, considering that they were related to the canopy and tree vigor description for both cultivars. All the other bibliographic references that we found about ‘Regina’ and ‘Sweetheart’ cultivars, described the characteristics of the fruits rather than canopy or vigor descriptions. That is the reason  why we did not used more references.

On the other hand. You ruled out factors like: exposure, microrelief, age of individuals, proximity to irrigation matrices (potentially leaking), or proximity to irrigation channels?

Answer: Yes, when we established the 'Regina' and 'Sweetheart' experiments, we checked the plots with the irrigation manager in the field. The topography is flat, and there are no problems with the irrigation system because we eliminated all the original irrigation lines and replaced them with new ones for the experimental plots to avoid problems of damaged or obstructed emitters, and any other uncontrolled factor. Also, we checked the irrigation mains and did not detect any leaks or irrigation ditches near the experiment. All the trees had the same age at the beginnings of the experiments. During the experiment, the vigor of ‘Sweetheart’ was always higher than ‘Regina’ (through LAI estimations, which were higher for ‘Regina’ and ‘Sweetheart’ sampled trees respectively (2.23±0.55 and 2.58±0.72 m2 m−2)  (line 147). These values evidenced a difference of near 19 % in the trends of vigor among them), presumably because of the effect of the occasional water table that was found later. A more rigorous comparison of canopy vigor between both cultivars is not possible at this stage since, we do not have more field data. As it was mentioned in M&M section, the canopy management and cultural practices were not part of this experiment, and they followed the regular management established by the commercial orchard. Just as an interesting note in this aspect, the field was anecdotally known as "Los Patos" ("the Ducks") because it was an old floodplain, which was drained, for agricultural purposes.

Answer: Thank you very much for your valuable suggestions.

Answer: The symbols for significance in Figures 3 and 4 has been modified to 0.01 ‘**’, and 0.05 ‘*’, as suggested.

Answer: In regards to the last query of Reviewer 2, the relevant response has been already presented in the first comment. Thank you very much again for all your valuable suggestions.
